# Incidence and Risk Factors for the Development of Stress Fractures in Military Recruits and Qualified Personnel: A Systematic Review

**DOI:** 10.3390/ijerph22111760

**Published:** 2025-11-20

**Authors:** Patrick G. Campbell, Rodney Pope, Vinicius Simas, Elisa F. D. Canetti, Benjamin Schram, Robin M. Orr

**Affiliations:** 1School of Behavioural and Health Sciences, Australian Catholic University, Brisbane, QLD 4014, Australia; 2Sports Performance, Recovery, Injury and New Technologies (SPRINT) Research Centre, Australian Catholic University, Brisbane, QLD 4014, Australia; 3National Centre for Veterans and Families, Australian Catholic University, Brisbane, QLD 4014, Australia; 4Tactical Research Unit, Bond University, Robina, QLD 4226, Australia; rpope@csu.edu.au (R.P.); vsimas@bond.edu.au (V.S.); bschram@bond.edu.au (B.S.); rorr@bond.edu.au (R.M.O.); 5School of Allied Health, Exercise and Sports Sciences, Charles Sturt University, Albury, NSW 2640, Australia; 6Faculty of Health Sciences and Medicine, Bond University, Robina, QLD 4226, Australia

**Keywords:** bone, army, navy, air-force, defence, trainees

## Abstract

Stress fractures are a major force preservation risk in military organisations. Although incidence rates and risk factors have been widely examined, a synthesis of high-quality evidence has been lacking. This review aimed to synthesise findings from studies examining stress fracture incidence and risk factors in military populations. The protocol was registered on the Open Science Framework and reported in accordance with PRISMA guidelines. Three databases were searched, and data on incidence, risk factors, and risk ratios were extracted. Study quality was assessed using Joanna Briggs Institute appraisal tools. Seventy studies were included. The incidence of stress fractures in military recruit/trainee populations was substantially higher than among qualified personnel (13.7–1713 vs. 2.7–56.9 per 1000 person-years). Fractures occurred most often in the tibia, fibula, and metatarsals. Higher-risk sub-populations included older and female personnel. Recruits/trainees faced additional risks, such as the following: consuming >10 alcoholic drinks per week; underweight BMI; beginning training without prior exercise of ≥3 sessions/week or ≥7 h/week in the previous year; low serum 25(OH)D levels; prior use of non-steroidal anti-inflammatory drugs; and the initial training stages with the greatest physical loading. Military personnel, particularly recruits, experience high stress fracture incidence, with physical workload and other risk factors contributing to elevated risk.

## 1. Introduction

Stress fractures are of considerable interest to military organisations due to the considerable burden on operational effectiveness, medical discharge, and economic costs [1,2,3,4]. For example, evidence demonstrates recruits who suffer stress fractures are over four-times more likely to be discharged than those who do not [2], and accounted for 29.7% of medical discharges from the U.S. Marine Corps [2]. This makes stress fractures a substantial force preservation risk to militaries that warrants close examination to determine factors that increase their likelihood of occurring, alongside prevention strategies. The aetiology of stress fractures and how this relates to known intrinsic and extrinsic risk factors in military populations is not yet entirely clear. However, stress fractures typically occur when bone is repeatedly exposed to sub-maximal load exposures, without adequate recovery (inclusive of nutrient restoration) to allow for bone remodelling processes to occur [5,6]. This situation is observed when excessive stress and strain is repetitively applied to specific bone regions (commonly, the tibia, metatarsals, and femoral neck); eventually the loading continuum will shift from adequate remodelling in response to mechanical load to bone stress injuries and eventually fracture [5,6,7,8]. This means the mechanical loading of biological tissue, represented through the mechanical load-tissue response pathway, is a prominent model for explaining critical failures of biological tissues (in this case, bone) [5,9].

The incidence of stress fractures in military recruits is substantial, with rates of up to 66.0 cases per 1000 person-years reported in U.S. armed forces recruits, compared to 1.9 (male) and 6.7 (female) cases per 1000 person-years in qualified personnel [10]. This disparity likely reflects the high volumes of unaccustomed marching, running, load carriage, callisthenics, and battle training involved in military training of new recruits [11]. These activities, often performed at higher intensities than recruits have experienced previously [12], increase their vulnerability to overuse-related syndromes and musculoskeletal injuries, including stress fractures (the latter because their bones are not well adapted for the training load) [11].

Risk factors for stress fractures discussed within the literature have been primarily investigated within various populations. For example, sex appears to be a prominent risk factor for development of stress fractures in sporting and military (both recruit and qualified personnel) populations [13,14]. The increased risk observed in females undertaking similar workloads to their male counterparts is likely multifactorial, and may be related to average differences between the sexes in bone mineral density, hormones, physiological factors, and training status (i.e., physical preparedness), among other potential factors [13,14]. Previous levels of physical activity and training status prior to basic training have also been prominently noted [6,15,16,17], alongside several biomechanical [15,18,19] and genetic factors [20,21]. There are also several pharmacological and nutrient-related risk factors, including non-steroidal anti-inflammatory drugs [22] and methylphenidate [23], vitamin D [24], inflammatory markers [25], calcium, and iron [26,27]. Together, these elements interact to create a multifactorial risk profile that varies between individuals.

Despite extensive research, there remains a surprising lack of systematic evidence synthesis regarding stress fracture incidence and risk factors across military populations. To the authors’ knowledge, there are currently no systematic reviews providing comprehensive evidence synthesis relating to incidence rates and risk factors for stress fractures within military personnel encompassing all service branches and both military recruits/trainees and qualified personnel. Crucially, no such reviews have examined the differences in the incidence and risk factors for stress fractures across service categories and between recruits and qualified personnel. Therefore, the aim of this systematic review was to identify and synthesise evidence reporting on incidence and risk factors for stress fractures within military recruits, trainees, and qualified personnel. Improved understanding of these factors may help military forces to maintain force readiness, operational effectiveness, and reduce medical discharge rates.

## 2. Materials and Methods

A systematic review was conducted to identify and synthesise findings from published studies that have investigated the incidence of stress fractures and occupational tasks and other factors associated with risk of stress fractures occurring in military populations. The reporting of this systematic review was guided by the recommendations of the Preferred Reporting Items for Systematic Review and Meta-Analysis (PRISMA) [28].

### 2.1. Protocol and Registration

The project and the protocol for this systematic review were registered with the Open Science Framework (OSF) registry on 27 July 2020 (accessible at https://osf.io/w6e7x/) [29]. Several alterations to the registered protocol were required, and these are detailed in an amendment to the OSF protocol and Appendix A. Briefly, the original review was to include all paid occupations and the decision was subsequently made to restrict this to solely military recruits or qualified personnel; and the original review comprised both traumatic and stress fractures, and due to restrictions in publication word limits, the overarching review was divided into two systematic reviews (i.e., one for each of traumatic fractures and stress fractures—the latter being the topic of this review).

### 2.2. Eligibility Criteria, Information Sources, and Search Terms

A systematic search of key databases was completed in May 2024. The databases PubMed, Elton B Stevens Company (EBSCO, including SPORTDiscus and Cumulative Index to Nursing and Health Care Literature [CINAHL]), and ProQuest (including Consuming Health Databases, Health and Medical Collection, Military Databases, Nursing and Allied Health Database, and Public Health Database) were searched using search terms derived from the following three themes: ‘fractures’, ‘work’, and ‘risk’. An example search string for the PubMed database can be found in Table 1 and the search strategies for the other databases can be found in Appendix A.

### 2.3. Inclusion and Exclusion Criteria

Specific inclusion and exclusion criteria are detailed in Table 2.

Additionally, articles were (a) excluded if they were qualitative research, (b) investigated populations with pre-existing medical conditions or defined based on attendance at a clinic or admission to hospital, (c) prospective studies in the elderly, (d) unpublished theses, (e) research protocols; or (f) were published prior to the year 2000.

### 2.4. Study Selection, Data Collection Process, and Data Items

All articles identified in the initial search were imported into reference management software (Endnote X9, version X9.3.3, Clarivate Analytics, Philadelphia, PA, USA), where duplicates were removed. Articles were then screened for relevance by title and abstract by one reviewer (PC), with those that were clearly not relevant to consideration of the incidence and risk factors for either fractures or stress fractures excluded at this stage. Remaining articles were obtained in full text for further assessment of eligibility against the eligibility criteria, to determine the final list of studies to be included in the review (PC and RP). Articles pertaining only to acute fractures and not to stress fractures were excluded at this stage. Disagreements on eligibility were resolved through discussion, and studies which were excluded at this stage were noted with reasons for exclusion and a list can be found in Appendix A. The results of the search, screening, and selection processes were documented in a PRISMA flow diagram [28].

Key data pertaining to study authors, study type, participant characteristics, methods of stress fracture identification and/or diagnosis, incidence or prevalence rates, and occupational exposures or risk factors assessed were then extracted and tabulated. Where possible, risk-related ratios, comparing levels of risks of developing fractures, were extracted, along with the associated 95% confidence intervals (CI). These included, for example, odds ratios (OR), relative risks (RR), hazard ratios (HR), and incidence rate ratios (IRR). Where raw data were presented in graph format, Web Plot Digitizer software (version 4) was used to extract the relevant data [30].

### 2.5. Levels of Evidence and Methodological Quality, Summary of Measures, and Synthesis of Results

The level of evidence provided by each study was evaluated according to the methods used by Robson et al. [31]. Each eligible study was assessed for risk of bias and critically appraised with specific tools for each type of study design, to assess their methodological quality (EC and PC). The Joanna Briggs Institute (JBI) critical appraisal tools for cohort studies, quasi-experimental and randomised controlled trials were used, as appropriate, to assess the trustworthiness, relevance, and results of included studies [32,33]. Questions asked in the corresponding checklists were answered as ‘yes’, ‘no’, or ‘unclear’. Due to this standardised approach, a scoring system was developed where one point was awarded for a ‘yes’ response and zero points were awarded for a ‘no’ or ‘unclear’ response. Methodological quality scores from these different tools were then assessed and qualitatively described using the approach implemented by Orr et al. [34]. A combination of the level of evidence and methodological quality assessment was used to classify, appraise, and grade the evidence presented by each study.

Key findings from the included studies were synthesised using a critical narrative approach. Meta-analysis of results from the included studies was not completed due to the heterogeneity across the studies included in this review.

## 3. Results

The systematic search resulted in the identification of a total of 24,093 studies (Figure 1). Following the removal of duplicates, articles that were identified as clearly not relevant during screening of titles and abstracts, and articles that were subsequently removed for specific reasons following review in full text to further assess eligibility, a total of 70 eligible articles were identified for inclusion.

Among the included stress fracture studies, there were 63 cohort studies, consisting of 33 that involved analysis of existing data sets [10,22,23,35,36,37,38,39,40,41,42,43,44,45,46,47,48,49,50,51,52,53,54,55,56,57,58,59,60,61,62,63,64,65], and 30 where data were collected prospectively [2,12,13,16,20,24,25,26,66,67,68,69,70,71,72,73,74,75,76,77,78,79,80,81,82,83,84,85,86,87]. These studies were considered to provide level III-2 and level II evidence, respectively [88]. Further, there were three studies that were randomised controlled trials [27,89,90], and four that were categorised as quasi-experimental [91,92,93,94], each considered to provide level II or III-2 evidence. The methodological limitations of the cohort studies primarily related to the potential for data coding errors and variability between coders to introduce data quality issues within the large, existing databases from which key data were extracted; a lack of specificity in exposure data relating to specific risk factors; use of differing historical control cohorts for statistical comparisons; and lack of consistency in approaches to dichotomising continuous variables between studies. The quasi-experimental studies were primarily limited by an inability to blind participants and/or individuals involved in the intervention, deviations from planned study protocols, or limited sample sizes completing the intervention arm. Methodological concerns relating to the randomised controlled trials primarily comprised potential complications with blinding of participants, a lack of detail regarding how blinding and randomisation occurred, and potential loss or bias introduced due to participant attrition and loss to follow-up.

All key data highlighting the occupational incidence rates, risk factors, study designs, methodological quality, and levels of evidence, extracted from the included studies, are tabulated in Appendix A. To address ease of readability and specificity of analysis, the data tables (and results synthesis) were divided into the following sections. Specifically, Appendix A details the characteristics and key findings of studies of military recruit/trainee populations, in relation to stress fractures; and Appendix A details the characteristics and key findings of studies of qualified military personnel and studies where recruit and qualified military populations could not be separated due to the time-periods of observation, in relation to stress fractures.

### 3.1. Occupational Incidence of Stress Fractures in Military Recruits/Trainees

A comparative summary of stress fracture incidence in military recruits or trainees by branch of service, nation, and sex is provided in Table 3. Briefly, the highest overall incidence rates for stress fractures generally occurred in cohorts of IDF recruits, with rates ranging from 123 to 309 cases per 1000 person-years for recruits in mixed-gender light combat units [25,26,81]; and ranging from 388 to 810 cases per 1000 person-years in cohorts of trainees or recruits in infantry and advanced combat training units [16,66,79,80]. Incidence rates of stress fractures as high as 1713 per 1000 person-years were reported for mixed-gender anti-aircraft IDF recruits [66].

In comparison, within the U.S. military there was a collective (whole of military) stress fracture incidence rate of 44.2 stress fractures per 1000 person-years [10], with rates ranging from 40.3 to 871.7 stress fractures per 1000 person-years in male or mixed-gender cohorts of the Army [39,64,77] and Marine Corps [50,84], and 29.2–383.5 cases of stress fracture per 1000 person-years in male or mixed-gender cohorts of the Army [60,68,72,73], Navy [27], and Air Force [89]. Similar stress fracture incidence rates were reported in U.K. military recruit training settings (85.2–202.3 stress fractures per 1000 person-years [40,57,65,70], and 62.6 cases per 1000 person-years) [12], Australian Army recruits (17 stress fractures per 1000 person-years) [49], Indian military recruits (87.3 cases per 1000 person-years) [69], and Finnish conscripts (99.0–116.0 cases per 1000 person-years) [24,83]. A higher stress fracture incidence rate closer to the rates observed within the IDF was noted in Chinese infantry recruits (878.8 cases per 1000 person-years) [20].

Stress fractures predominantly affected the lower limbs, with the tibia, fibula, and metatarsal bones being the most commonly observed fracture sites across militaries around the world. Table 4 provides a detailed synthesis of stress fracture incidence rates in military recruits/trainees by anatomical location and split by population.

### 3.2. Occupational Incidence of Stress Fractures in Qualified Military Personnel

Qualified military personnel, though at lower risk than recruits, still exhibit notable stress fracture incidence rates. In active-duty U.S. Army soldiers (2001–2011), the incidence rate was 4.12 per 1000 person-years, while across all active-duty U.S. military members (2003–2012) [35], it was 2.7 per 1000 person-years, dropping to 0.7 in deployed personnel [10]. In these deployed personnel, the tibia/fibula was the most affected site (1.2 per 1000 person-years), followed by metatarsals and other bones [10]. Conversely, Waterman et al. [56] reported a slightly higher rate of 5.69 lower limb stress fractures per 1000 person-years in the U.S. armed forces (2009–2012), likely due to the inclusion of recruits. Consistent with other studies discussed above, stress fractures most commonly affected the tibia (2.26 per 1000 person-years) and least often affected the femoral shaft (0.34 per 1000 person-years) [56].

Waterman and colleagues [56] also observed the highest incidence rates for lower limb stress fracture were in the Army and Marine Corps, particularly among junior enlisted personnel, where rates reached 18.54 per 1000 person-years, compared to 3.89 for junior officers [56]. Junior enlisted personnel also had substantially higher adjusted incidence rates for total lower limb stress fractures, femoral neck stress fractures, and tibial stress fractures when compared to senior enlisted personnel and junior and senior officers [56]. These disparities were even higher for femoral neck and tibial stress fractures (29.76 and 30.76 stress fractures, respectively, in junior enlisted personnel, compared to 3.04 and 5.63 stress fractures, respectively, in junior officers, per 1000 person-years) [56]. These findings are also consistent with the incidence rate observed in U.S. Army soldiers for the years 1996–1997, with the total incidence in this population reported as 21.6 stress fractures per 1000 person-years [82]. Finally, in IDF personnel (for all combat soldiers, which is inclusive of commencing recruits and qualified personnel) the overall rate for qualified personnel was comparatively higher (56.9 per 1000 person-years) than that for U.S. qualified personnel, especially at the tibia/fibula (46.2 per 1000 person-years) [23]. The exception to qualified personnel being at lower risk of stress fracture when compared to recruit personnel was a cohort of U.S. Air Force Special Warfare trainees, who experienced 164.7 cases of stress fracture per 1000 person-years [61].

### 3.3. Occupational Tasks and Injury Mechanisms Associated with Stress Fractures in Military Recruit or Trainee Populations

Pihlajamaki et al. [51] found the majority of stress fractures contributing to the cumulative incidence of stress fractures occurred during the first 4 months of Finnish military service, with fracture incidence rates plateauing after this point, up to the concluding 12-month time-point. Dash et al. [69] found a similar effect in Indian military recruits, with almost 80% of the stress fractures they observed occurring during the initial stages of basic training. These findings are consistent with those of Mattila et al. [78], who found a lower overall rate of occurrence of stress fractures in male and female Finnish military conscripts serving more than 362 days when compared to those serving 180 days (HR 0.7, 95% CI 0.5–0.9).

A week-by-week analysis of stress fracture incidence in U.S. Army soldiers during the initial 6 months of service was provided by Kardouni et al. [64]. Incidence rates peaked between weeks 4 and 8 (1.57–1.74 stress fractures per 1000 soldiers per week), with the highest incidence occurring in weeks 5 (1.74 stress fractures per 1000 soldiers per week) and week 8 (1.74 stress fractures per 1000 soldiers per week) [64]. Incidence then progressively decreased from week 9 (1.46 stress fractures per 1000 soldiers per week) to week 26 (0.08 stress fractures per 1000 soldiers per week). When stratified by sex, the incidence of stress fractures peaked at week 5 for males (1.23) and at week 7 for females (4.54), with incidence progressively declining as training and service continued through to week 26 [64]. Importantly, the incidence rate in female soldiers for stress fractures was substantially higher than that in male soldiers, throughout the entire initial 26-week period [64].

Moran and colleagues [79] investigated associations between training volume and stress fracture incidence. Their data indicated a significantly higher stress fracture incidence in male IDF recruits who had higher average daily step counts (mean ± SD 16,276 ± 3317; measured via a pedometer) than in recruits with lower counts (mean ± SD 14,103 ± 2302) [79]. However, while the observational data from that study [79] supported an association between external training load (i.e., training volume) and stress fracture incidence, a separate quasi-experimental trial did not find any clear associations between manipulated external training load variables and stress fracture incidence [92]. In considering the findings of that quasi-experimental study, however, it should be noted the allocation of training regimes occurred at a company level rather than individual level and was non-random, and so confounding by other variables associated with each company may have occurred and obscured any actual effects of training loads or regimes.

Another study, conducted in IDF infantry units across the years 1983–2015 [46], examined changes in stress fracture incidence rates that accompanied changes in training regimes across the years. That study again suffered from inadequately controlled risk of confounding due to changes in training contexts and cohorts that may have occurred over time, alongside the changes in training regimes. Noting this limitation, the study observed stress fracture incidence rates were not significantly affected by moving training to flatter terrain, reducing marching and formal running distance, or increasing lower body strengthening exercise as a component of the overall training regime [46]. However, stress fracture incidence was significantly reduced when training was restricted to authorised protocols only (platoon commanders were adding additional loading to authorised conditioning programmes); comparing stress fracture risk when training was not restricted when it was restricted to authorised protocols only—for radiographically verified stress fractures, the OR was 3.17 (95% CI 1.10–9.09), and for radiographically and clinically verified stress fractures, the OR was 3.87 (95% CI 1.52–9.83) [46]. These results suggest an association may exist between training volume and incidence of stress fracture.

In female IDF recruits, two studies reported an increased incidence of stress fractures when recruits carried higher loads [67,90]. However, in one of these studies [90] the difference in incidence rates was minimal and in the second study [67], an error in calculation of one incidence rate was detected, which likely means the reported difference in incidence rates was less than reported and in the opposite direction. In the first study, implementing a new fighting vest specifically designed for women resulted in a slightly increased incidence of stress fracture (389 vs. 386 cases per 1000 recruit-years) [90]. While the difference was minimal, the new fighting vest designed for women weighed more (weight = 1950 g) than the standard fighting vest (weight = 1350 g) and that heavier weight may have been one reason why stress fracture rates did not decrease with the new female vest [90]. The second study [67] reported that training with heavy equipment increased the odds of recruits sustaining stress fractures (OR 4.0, 95% CI, 2.1–7.6). However, the OR reported by the authors appears to be both excessive and inverse to the true OR, given the calculation errors we have highlighted for that study (see Appendix A), and as such we suggest these findings be disregarded. On that basis, there is no robust evidence available from the included studies to indicate heavier load carriage is associated with higher stress fracture incidence rates.

Of concern for defence agencies, in male U.S. Marine Corps recruits, Reis et al. [2] demonstrated the detrimental effect of stress fractures during training, with occurrence of stress fracture being associated with substantially increased risk of being discharged from military service (OR 4.19, 95% CI 2.73–6.45).

### 3.4. Occupational Tasks and Injury Mechanisms Associated with Stress Fractures in Qualified Military Personnel

No studies reported on specific occupational tasks or injury mechanisms associated with stress fracture incidence in qualified personnel.

### 3.5. Other Factors Associated with Stress Fractures in Military Recruit or Trainee Populations

#### 3.5.1. Health or Medical History

History of prior injury (compared to no history) was a substantial risk factor for stress fractures in both male (RR 3.58, 95% CI 1.13–11.34) and female (RR 6.06, 95% CI 3.02–12.14) U.S. military academy cadets [75]. Risk was further exacerbated if the previous injury resulted in activity limitation, with male cadets (RR 17.03, 95% CI 4.73–61.29) and female cadets (RR 9.68, 95% CI 3.91–23.95) who had such a history having substantially increased risks of developing a stress fracture when compared to those without [75]. It should be noted the confidence intervals of these relative risk estimates are very wide, indicating considerable uncertainty regarding their magnitude [75]. Similar findings were noted in male Chinese infantry recruits, where recruits with a history of prior stress fracture were at greater risk than recruits with no prior history (OR 1.77, 95% CI 1.13–2.77) [20].

Both history of smoking (RR 1.34, 95% CI 1.05–1.71) and years smoked (RR 1.05, 95% CI 1.02–1.08) were associated with stress fracture risk in one study [77]. However, while three other studies also found stress fracture risk to be higher in smokers than in non-smokers (IRR 1.41 (95% CI, 0.91–2.21) [36], RR 1.31 (95% CI, 0.99–1.75) [27], and RR 1.22 (95% CI 0.89–1.67) [76]), the 95% confidence intervals around the three resulting estimates of relative risk each crossed 1.00; thus, including risk estimates which indicate history of smoking may be associated with either a heightened or reduced stress fracture injury risk. Zhao and colleagues [20] also investigated smoking status and risk of stress fracture and similarly found no statistically significant association.

Presence of diagnosed anaemia in IDF military personnel was associated with significantly (though not substantially) increased odds of stress fracture occurrence (OR 1.04, 95% CI 1.02–1.07) [23], and an increased incidence of stress fractures was also noted in personnel with anaemia at baseline and following the completion of basic training [26]. Female Navy and Marine Corps recruits who had diagnosed amenorrhea (OR 1.86, 95% CI 1.41–2.45) [27] or reported zero menses during the past year (OR 5.85, 95% CI 1.7–20.8) [84] had increased odds of sustaining stress fractures. However, the positive association between use of depot medroxyprogesterone acetate (vs. non-use) and stress fracture risk (OR 1.24, 95% CI 0.91–1.68) did not reach statistical significance [27]. Finally, recruits with diagnosed and treated ADHD had marginally higher odds of sustaining stress fractures (OR 1.04, 95% CI 1.02–1.07) than those with untreated ADHD (OR 1.01, 95% CI 0.99–1.03) or in a control group (no ADHD diagnosis) (OR 1.00, reference) [23]. However, the magnitudes of these associations were very small.

#### 3.5.2. Sex

The incidence of stress fractures in female compared to male military recruits was substantially higher, regardless of branch of service or nation (e.g., United States, Finland, Australian, Israel, etc.) [10,39,48,49,66,68,72,93,95]. The incidence rates and incidence rate ratios, where available, are displayed in Table 3, separated by nation and service branch where possible. Briefly, incidence rate ratios indicated stress fracture risk was two to four times higher in female than in male U.S. Army recruits [39,48,64]. These findings in U.S. populations were consistent with those in IDF, Australian, and Finnish recruit populations (Table 3).

In a large-scale cohort of U.S. Army recruits (*n* = 583,651), the odds of female recruits sustaining stress fractures during basic training were 3.83 (95% CI 3.66–4.05) times higher than the odds for male recruits [44]. Montain et al. [47] also found female U.S. Army basic combat trainees had substantially higher odds of stress fracture occurrence than male trainees (OR 4.51, 95% CI 4.36–4.66). A similar ratio of stress fracture risks was found in a cohort of Finnish military conscripts, with female compared to male recruits having a hazard ratio of 8.2 (95% CI 4.8–14.2) [78].

In contrast, among U.S. Marine Corps *officer candidates* completing basic training, stress fracture incidence was higher in male candidates (0.05 stress fractures per 1000 training-hours) than female candidates (0.03 stress fractures per 1000 training-hours) [50]. These officer candidate incidence figures provide a female/male incidence rate ratio of 0.6, which is an outlier among the included studies of military personnel undertaking basic training within this review. This finding suggests Marine Corps officer candidates undertaking basic training may have completed a variation on conventional military basic training, or alternatively, that preceding stages of officer candidate selection or sex-based differences in basic training programming may have resulted in the lower rates of stress fracture occurrence in female candidates.

Kardouni and colleagues [64] also noted, when examining the distributions of stress fractures by anatomical location in male and female soldiers undertaking their first 6 months of military service, that there were substantial differences between the sexes in the proportions of stress fractures affecting the tibia/fibula (males 34.9%, females 19.2%), metatarsals (males 13.8%, females 5.2%), and pelvis (males 3.7%, females 16.9%) [64]. These findings indicate some sex-based differences in anatomical areas susceptible to stress fracture.

#### 3.5.3. Age

Overall, evidence from included studies indicates risk of stress fracture increases progressively with recruit age in men and women, regardless of service branch or nation [10,27,36,44,55,77,78]. For example, in a large-scale study of U.S. Army recruits the odds of sustaining stress fractures progressively increased as recruit age progressed from <20 years (OR 1.00 [male and female reference groups]) to 20–24 (♂ OR 1.47, ♀ OR 1.41), 25–29 (♂ OR 2.33, ♀ OR 1.80), and ≥30 years (♂ OR 3.50, ♀ OR 2.29) [44]. Interestingly, this association was more pronounced in male compared to female recruits, although the greater incidence rate overall in female recruits should be noted when considering this latter finding [44]. Similar findings were identified in male and female Finnish military conscripts, with hazard ratios for recruits aged 17–19 years (HR 1.00 [reference group]) lower than those for recruits aged 20 years (HR 1.2, 95% CI 0.9–1.7), and 21–29 years (HR 2.1, 95% CI 1.4–3.1) [78]. In cohorts of U.S. female service-members, incidence rate ratios also progressively increased as age categories increased from 18 to 19 years (IRR 1.00 [reference group]) to 20–24 years (IRR 2.06, 95% CI 1.32–3.20) and ≥25 years (IRR 3.07, 95% CI 1.81–5.19) [36]. Finally, female naval recruits who were ≥25 years of age had greater odds of stress fractures than younger recruits (OR 1.65, 95% CI 1.07–2.45) and risk of stress fracture occurrence in female army recruits increased significantly with each additional year of age (RR 1.07, 95% CI 1.05–1.10) [27,77]. In sharp contrast, there was one study conducted within an IDF cohort which indicated a small reduction in the odds of sustaining a stress fracture with each additional year in age (OR 0.994, 95% CI 0.991–0.997) [23]. Sormaala and colleagues [55] also identified that there were no significant associations between stress fracture risk and age (or sex, length of service, aerobic fitness level, or muscular strength) for *talus* stress fractures (in the hindfoot).

#### 3.5.4. Body Composition

Several of the included studies identified associations between body composition measures (e.g., BMI or body fat percent) and risk of stress fractures [36,44,45]. The clearest findings related to increased risk of stress fracture in the ‘underweight’ BMI category of <18.5 kg/m^2^ when compared to the ‘normal’ category (i.e., 18.5–24.9 kg/m^2^) in male (OR 1.78, 95% CI 1.60–1.98) and female (OR 1.31, 95% CI 1.19–1.45) personnel within a large U.S. Army recruit population (*n* = 583,651) [44]. A similar finding for the underweight category of BMI was also separately identified in female U.S. Army members, with underweight personnel being at increased risk when compared to those in the ‘normal’ category (IRR 2.63, 9%% CI 1.38–5.02) [36]. Conversely, Mattila and colleagues [78] did not find any association between BMI category and risk of stress fractures in Finnish conscripts when using a different system for categorising BMI, such that the lowest BMI category was defined as BMI < 20 kg/m^2^ rather than <18.5 kg/m^2^. However, it is important to consider the likelihood of a clinically significant difference between the BMI category of <20 kg/m^2^ used in that study and the underweight category of <18.5 kg/m^2^ used in other studies, which might have been the reason for this null finding. Additionally, in one study, female U.S. Army recruits in the ≥25 kg/m^2^ category (‘overweight’ or ‘obese’ categories) had significantly lower rates of stress fractures than those in the ‘normal’ and ‘underweight’ categories (OR 0.87, 95% CI 0.83–0.92 for those overweight and OR 0.82, 95% CI 0.68–0.99 for those obese when compared to the ‘normal’ category) [44]. However, further evidence of this negative (protective) relationship between overweight or obesity and stress fracture risk was not provided by other studies.

Zhao and colleagues [20] identified no statistically significant associations between age, height, body weight, or leg length and incidence of stress fractures in Chinese military recruits. It is unsurprising there were also no clear associations identified in other studies of the links between body composition and stress fracture risk where a dichotomised categorisation of body composition was employed, which broadly categorised recruits as either weight-qualified or having excess body fat [37,74]. Use of such dichotomised systems of categorisation based on body composition lacks the specificity and range of rankings needed to provide optimal statistical power to detect clinically meaningful associations.

#### 3.5.5. Race and Ethnicity

The most consistent relationship identified within the U.S military recruit and trainee populations was that military recruits categorised as being of black race/ethnicity were observed to be at lowest risk of stress fracture, when compared to those categorised as being of white, Hispanic, Asian, American Indian, ‘other’, or ‘unknown’ race/ethnicity [44,45,47,77].

#### 3.5.6. Prior Physical Activity Levels

Frequency and hours per week of previous exercise were associated with risk of stress fracture in military recruit and trainee populations [20,27,51,68,77,80]. Male U.S. military cadets exposed to less than 7 h per week of exercise in the prior 12 months were found to have a greater risk of stress fracture than cadets exposed to more than 7 h per week (RR 2.31, 95% CI 1.29–4.12) [68]. This finding was replicated in male Chinese infantry recruits, using identical categories for the risk factor (OR 1.84, 95% CI 1.32–2.56) [20]. However, the association was not evident in female U.S. military cadets, in a study again using identical categories for this risk factor (RR 1.27, 95% CI 0.60–2.74) [68]. Conversely, when exercise history was categorised using frequency of prior exercise activity (rather than hours per week), frequencies ≥ 3 times/week were associated with a substantially lower risk of stress fracture development in female U.S. Army recruits in two different studies (OR 0.66, 95% CI 0.51–0.85 [27] and RR 0.65, 95% CI 0.51–0.82 [77]). Prior exercise history was also evaluated in Finnish male conscripts, with those recruits completing ≥2 exercise sessions per week at an intensity eliciting breathlessness having a reduced incidence rate for stress fractures when compared to those who did not (IRR 0.41, 95% CI 0.20–0.85) [51].

Evidence for reduced stress fracture risk in recruits demonstrating or recalling a significant history of exercise was also provided by two studies of IDF army recruits [16,80]. Firstly, recruits with a history of participating in ball sports had significantly reduced odds of sustaining stress fractures when compared to recruits with no history of playing ball sports, in three separate cohorts from different time periods (specifically, 1998 cohort [OR 0.37, 95% CI 0.21–0.66], 1990 cohort [OR 0.54, 95% CI 0.29–0.99], 1995 cohort [OR 0.16, 95% CI 0.03–0.69]) [16]. Secondly, Moran and colleagues [80] developed a predictive model examining the interplay between prior aerobic training (times per week) and prior aerobic training volume (minutes per session) in predicting stress fracture risk in male army recruits [80]. The model demonstrated stress fracture risk was lower when frequency of prior aerobic training (sessions per week) was higher (OR 0.22, 95% CI 0.08–0.59), and higher when aerobic training sessions were longer in duration (OR 1.66, 95% CI 1.02–2.69). Importantly, these aerobic training variables were the largest contributors to the prediction model, which was able to successfully predict presence or absence of stress fractures in 76.5% of recruits within a validation cohort. In addition, male (OR 0.68, 95% CI 0.57–0.80) and female (OR 0.60, 95% CI 0.43–0.82) recruits who reported a high subjective rating of physical conditioning within the Tailored Adaptive Personality Assessment System (TAPAS) at time of entry into military service had reduced odds of sustaining stress fractures during basic training [48].

#### 3.5.7. Aerobic or Strength Test Performance

Relationships between performance outcomes for tests of aerobic capacity or muscular strength in military recruits and subsequent risk or incidence of stress fractures are less clear. The relationships between aerobic fitness test performance and stress fracture risk in U.S. female naval recruits in one study [27] and male and female Finnish conscripts in others [24,78,96] were unremarkable, with 95% confidence intervals for odds ratios and hazard ratios crossing one. Similar findings were demonstrated between muscular strength performance and stress fracture risk [24,78]. Additionally, no statistically significant associations between army physical fitness test results and stress fracture risk were identified in male Chinese recruits [20].

The clearest identified association was that the relative risk of stress fracture increased by a factor of 1.35 (95% CI 1.25–1.45) for every minute a recruit ran slower than the median 1.5 mile run time for the overall cohort [27]. Further, when female Marine Corps recruits’ 1-mile timed-run performances were categorised in quartiles; recruits with times which fell into the slowest quartiles, Q3 (OR 3.89, 95% CI 1.6–9.6) and Q4 (OR 3.14, 95% CI 1.2–9.0), had significantly higher odds of sustaining stress fractures than those with a Q1 run time (OR = 1.00, reference) [84]. However, the fact the point estimate of the odds ratio for Q4 was not higher than the point estimate for Q3 suggests there might not be a clear dose–response relationship between aerobic fitness level and stress fracture risk—there might instead be a critical threshold level of fitness that is associated with reduced risk. Supporting this possibility, Cowan and colleagues [36] identified female army recruits who failed an aerobic step test at the commencement of basic training had a higher incidence of stress fracture than those who passed (IRR 1.76, 95% CI 1.18–2.63). This finding was replicated by Krauss and colleagues [74] in the same population, with unfit female recruits found to have a significantly increased risk of stress fracture (IRR 1.62, 95% CI 1.19–2.21).

#### 3.5.8. Pharmacological Factors and Blood-Markers

A variety of pharmacological factors and blood-based biomarker risk factors for stress fracture were considered within the literature, but few were significantly associated with risk of stress fracture. For example, no clear associations were found between stress fracture risk and baseline parathyroid hormone levels [70], TRACP-5b (i.e., serum tartrate acid phosphatase isoform 5b) levels taken at regular intervals [83], or deoxypyridinoline (i.e., a biomarker of bone resorption) levels taken at regular intervals [93]. However, in a randomised controlled trial of calcium and vitamin D supplementation, a significantly lower proportion of the intervention group (calcium + vitamin D) (5.3%) than of the control group (6.6%) was diagnosed with stress fractures, with an RR of 0.80 (95% CI 0.64–0.99) [27]. Another study also reported significant associations between 25(OH)D levels and stress fracture risk; concentrations of 25(OH)D less than 50 nmol L^−1^ in Royal Marines recruits during week 1 of training were associated with increased odds of sustaining a stress fracture when compared with 25(OH)D concentrations above these levels (OR = 1.6, 95% CI 1.0–2.6) [70]. Similar results were noted in a cohort of male Finnish infantry recruits; however, in that study 25(OH)D levels less than 75.8 nM were associated with greater odds of stress fractures when compared to levels greater than 75.8 nM [24]. In U.S. Navy recruits who developed stress fractures, those who were categorised as vitamin D deficient (≤30 ng/mL) had higher physical therapy treatment costs than recruits with low–normal levels (31–40 ng/mL) (*p* = 0.049) [60]. Together, these findings suggest vitamin D levels may be inversely associated with stress fracture risk and that vitamin D supplementation may be protective.

Other variables which have been associated with stress fracture risk include intake of non-steroidal anti-inflammatory drugs (NSAIDs) [42] and levels of iron [25,81], transferrin [25], and ferritin [81]. For example, the incidence rate ratio comparing stress fracture incidence for U.S. Army recruits with a history of ingesting NSAIDs to those with no history was 5.3 (95% CI 4.9–5.7) [42]. The incidence rate ratios were substantially greater than 1.0 across all identified types of NSAIDs investigated within that study (e.g., ibuprofen, naproxen, indomethacin, and paracetamol) in both the full Army population and basic training population [42]. These findings suggest that NSAIDs or paracetamol taken to treat musculoskeletal and non-musculoskeletal pain may increase the risk of stress fractures. However, it should also be considered that the underlying health condition that led to NSAID use might have been a key contributor to the observed increase in stress fracture risk that follows NSAID or paracetamol use. The study did consider this possibility, and when only NSAID use *unrelated* to musculoskeletal symptoms was considered, the incidence rate ratios comparing stress fracture risk in those with NSAID use to the risk in those without NSAID use were lower, though still significantly greater than 1.0 [42]. This latter finding suggests NSAID use may play a role in increasing stress fracture risk, though prior injury (or experiencing the initial stages of bone stress injury) might also contribute.

In studies of IDF recruits, Moran et al. [81] used a logistic prediction model and found recruits with higher ferritin levels (aOR 1.03 for each single unit increase in ferritin levels, 95% CI 1.005–1.067) or lower iron levels (aOR 0.98 for each single unit increase in iron levels, 95% CI 0.97–1.0) had increased odds of sustaining stress fractures. Merkel and colleagues [25] did not report significant associations between stress fracture occurrence and ferritin levels in IDF recruits. However, lower iron levels at baseline (*p* = 0.02) and at 4 months of training (*p* = 0.004) were significantly associated with stress fracture occurrence [25]. In the same study, recruits who sustained stress fractures had significantly higher transferrin levels at baseline (*p* = 0.04) and at 4 months of training (*p* = 0.01) than those who remained injury-free [25].

#### 3.5.9. Biomechanical Factors

Dixon and colleagues [71] identified that for each one percent increase in the dynamic arch index (indicating a flatter or more flexible foot during movement) the risk of 2nd metatarsal stress fracture significantly reduced (RR 0.75, 95% CI 0.63–0.89), and for every one degree increase in foot abduction range of motion (again indicating a more flexible foot), the risk of 2nd metatarsal stress fracture similarly reduced (RR 0.87, 95% CI 0.80–0.96) during a barefoot running task. The same study also found that for each 1% increase in time to peak pressure at the 2nd metatarsal, as a proportion of total stance time (indicating a stiffer or less flexible foot structure), the risk of stress fracture occurrence at the 3rd metatarsal increased significantly (RR 1.19, 95% CI 1.04–1.35) [71]. In a separate study, greater time to maximum foot pronation (indicating foot pronation continued for longer and so the foot was more flexible) was found to be associated with lower risks of stress fracture at the femur and tibia during treadmill walking [73] in IDF recruits. Other factors considered in that study, including maximal pronation angles, range of motion, measured time from heel strike to pronation, and angular velocity of foot pronation, were not significantly associated with stress fracture risk [73]. Finally, using shock-absorbing insoles (SAI) in boots of Royal Marine recruits completing 32 weeks of commando training resulted in a substantial reduction in stress fracture incidence rates when compared to use of the conventional Saran insole (comparing Saran insole to SAI, OR = 1.71, 95% CI 1.21–2.42) [41]. The most substantial reductions in stress fracture risk were identified at the tibia (comparing Saran insole to SAI, OR = 3.19, 95% CI 1.36–7.45) [41]. Taken together, these studies suggest greater shock absorption at the foot/ground interface, achieved by use of shock-absorbing insoles or greater foot flexibility and pronation, may lower the risk of metatarsal, femoral, and tibial stress fractures.

#### 3.5.10. Genetic Markers

Findings of a study in Chinese infantry recruits indicated increased odds of those with, when compared to those without, stress fracture having greater GDF5rs143383 T (rather than C) allele frequency (OR 1.75, 95% CI 1.35–2.28), and greater frequencies of the following genotypes: codominant TT (vs. CC and TC) (OR 1.76, 95% CI 1.29–2.38), dominant TT + TC (vs. CC) (OR 2.91, 95% CI 1.25–6.74), and recessive TT (vs. CC + TC) (OR 1.83, 95% CI 1.33–2.52) [20].

#### 3.5.11. Other Factors

Other factors identified in the included studies to be associated with stress fracture risk in military recruits or trainees included UV index of the recruit’s home residence, height (stature), years from menarche (female recruits), characteristics or biomarkers of skeletal bone, alcohol consumption, and self-reported burnout [47,68,76,77,78,93,97]. Contrary to our expectation based on evidence regarding the relationship between vitamin D levels and stress fracture risk (see above), recruits whose home place of residence was classified as having a low UV index were at slightly lower risk of stress fractures—both women (OR 0.89, 95% CI 0.84–0.95) and men (OR 0.92, 95% CI 0.87–0.97)—than recruits from areas with a high UV index (OR 1.00, reference group) [47]. There was an unclear relationship between height (stature) and stress fracture risk in Finnish military conscripts [78], when assessed using categorisation of recruits by quartiles of height; however, in IDF army recruits, there was evidence of a significant increase in odds of stress fracture with each one centimetre increase in height (OR 1.08, 95% CI 1.01–1.15) [81].

Some specific skeletal bone characteristics were also associated with stress fracture risk. Each one-millimetre reduction in diameter of the femoral neck was associated with a significant increase in the risk of stress fracture occurrence in United States Military Academy (USMA) female cadets (RR 1.16, 95% CI 1.01–1.33) [68]. In USMA male cadets, each 10 mg reduction in tibial bone mineral content was similarly associated with a significant increase in the risk of stress fracture occurrence (RR 1.11, 95% CI 1.03–1.20) [68]. A study that considered findings from quantitative ultrasound also identified significant increases in relative risk of stress fracture occurrence to be associated with single unit of measurement decreases in speed of sound (SOS) and broadband ultrasound attenuation (BUS) in the calcaneal bone—each signifying a reduced bone mineral density—in female U.S. Army recruits [76]. For undisplaced femoral neck stress fractures, specifically, risk factors including BMI, bone neck-shaft angle, or leg dominance were not found to be significantly associated with stress fracture risk in a study of more than 500,000 recruits [52]. Similarly, levels of deoxypyridinoline (a biomarker of bone resorption) were not found to be significantly associated with incidence of stress fractures in Marine Corps recruits; however, the authors of that study noted it was underpowered to detect any clinically significant association [93].

Consumption of greater than ten alcoholic drinks per week was associated with a threefold increase in the risk of stress fracture occurrence in female U.S. Army recruits (RR 3.22, 95% CI 1.82–5.69) [77]—a finding that was replicated by Lappe et al. [76] in the same population (RR 3.08, 95% CI 1.59–5.97). Other examined interventions involved the use of an athletic trainer to provide care to a group of U.S. Air Force recruits, which resulted in reduced rates of stress fracture (rate ratio 0.84, 95% CI 0.73–0.97) [89].

Finally, fewer years since menarche were associated with increased stress fracture risk in USMA female cadets (RR 1.44 for each year less since menarche, 95% CI 1.19–1.73) [68]; and male and female IDF army recruits who self-reported psychological ‘burnout’ had increased odds of sustaining stress fractures when compared to those who did not (OR 1.59 for each one-point increase on a seven-point scale indicating level of perceived burnout, 95% CI 1.04–2.41) [81].

### 3.6. Other Factors Associated with Stress Fractures in Qualified Military Personnel

#### 3.6.1. Sex

An increased incidence of stress fractures was observed among qualified female U.S. and Israeli military personnel when compared to male personnel [23,35,56,61,82]. For example, Potter et al. [82] reported qualified female U.S. Army soldiers experienced more than double the incidence of stress fractures (i.e., 50.4 stress fractures per 1000 person-years) experienced by males (20.4 stress fractures per 1000 person-years). This equates to a female/male incidence rate ratio for stress fractures of 2.47 [82]. Similarly, within cohorts of qualified U.S. military personnel and Air Force special warfare trainees female/male incidence rate ratios of 3.63 and 3.11 were reported for stress fractures [35,61]. Additionally, Waterman and colleagues [56] found there were three-fold greater odds of stress fracture (OR 3.11, 95% CI 3.03–3.18) in female compared to male (1.00, reference rate) qualified U.S. personnel (adjusted for service branch, race, age, and rank).

#### 3.6.2. Race/Ethnicity

Estimates of adjusted incidence rates for stress fracture by race category (white, black, or other) indicated ‘white’ U.S. military service members had the highest overall, femoral neck, and tibial stress fracture incidence rates, when compared to personnel categorised as being of ‘black’ (reference category) and ‘other’ race [56]. Bulathsinhala et al. [35] employed more specific race-origin categories, and identified that service members categorised as being of ‘mixed-races’ (5.41 stress fractures per 1000 person-years) and ‘American Indian/Native Alaskan’ (5.11 stress fractures per 1000 person-years) had the highest incidence rates for stress fractures, while those categorised as ‘non-Hispanic black’ had the lowest incidence rate (2.72 stress fractures per 1000 person-years). This study also calculated adjusted hazard ratios for stress fracture occurrence, with the ‘non-Hispanic black’ race category constituting the reference category (HR = 1.00), and identified ‘non-Hispanic white’ male personnel (HR = 1.59, 95% CI 1.49–1.69) and female personnel (HR = 1.92, 95% CI 1.81–2.03) as the categories with the highest adjusted HRs among the included race-origin categories [35].

#### 3.6.3. Age

Within the U.S. armed forces, adjusted incidence rates for stress fractures, by age category, indicated the 20–24-year age group (which was the reference group) had the lowest incidence of overall, femoral neck, and tibial stress fractures [56]. The highest incidence rates for stress fractures were observed in the <20-year age group and within the ≥40-year age-group [56]. The results also demonstrated the incidence rates of stress fractures progressively increased in groups older than the 25–29-year age group and there was a substantial jump in incidence rate in the ≥40-year age group [56]. These findings were consistent across all of the evaluated anatomical areas (e.g., femoral neck and tibial), as well as for overall stress fracture rates [56]. For example, the adjusted incidence rate ratio for all stress fractures in those aged < 20 years was 3.14 (95% CI, 3.05–3.23) and in the ≥40-year age group was 6.4 (95% CI, 6.12–6.70), and adjusted incidence rate ratios for the 25–29, 30–34, and 35–39-year age groups ranged from 1.15 to 2.01 per 1000 person-years [56].

## 4. Discussion

The aim of this systematic review was to identify and synthesise current evidence regarding the incidence and risk factors for stress fractures in military populations. While there was a substantial volume of prospective cohort-based studies that evaluated *incidence* of stress fractures in military populations, there was less evidence from high level study designs which assessed factors associated with *risk* of stress fracture occurrence. Consequently, potential for confounding and inadequate consideration of additional explanatory variables was substantial and limited the ability to draw strong conclusions regarding specific exposures to risk factors and interventions that may be associated with stress fracture occurrence in military personnel.

The primary finding was that military recruits experience high incidence rates of stress fracture, regardless of nationality or branch or service, and these rates suggest the risk of developing stress fractures is greater in military initial training settings is greater than in physically active people from the general population [16,80,98,99]. The review also identified numerous extrinsic and intrinsic risk factors specific to military personnel, or arising from the occupational tasks of military forces, which appear to be associated in at least some contexts with the development of stress fractures. For example, identified risk factors included previous stress fracture, older age, female sex, being an enlisted service member, serving in the Army or Marine Corps services, some specific military physical training factors (particularly unplanned additional training loads), inadequate prior exercise history, specific medical conditions, and pharmacological, genetic, or blood-based biomarkers (see Appendix A). Evidence synthesised in this review may be of substantial importance in the risk management of stress fractures in military populations.

### 4.1. Stress Fracture Incidence

Stress fracture incidence rates in military recruit populations represented a substantial injury burden, regardless of nationality or service branch, although reported incidence rates varied substantially between nations. It is unclear what specifically causes such wide variation in the incidence of stress fractures. However, given the complex and multifactorial nature of injury aetiology and the substantial volume of evidence, covering multiple nations, this finding is unsurprising. The variability in reported incidence of stress fractures is likely to reflect a combination of factors which vary with context, including personal and/or anatomical characteristics of cohorts, differences in equipment and/or body armour design, self-selection biases in nations other than those with military conscripts (i.e., Israel and Finland), prevailing attitudes towards injury prevention, contextual variables which may affect injury rates or risk factors (i.e., environmental temperature, terrain, altitude, staffing), ratios of the sexes in cohorts, differences in training programme design, and differences in diagnosis and recording of stress fractures.

In a clear example of the influence of certain variables, the incidence of stress fractures in female U.S. Army recruits was substantially higher than that typically observed in male cohorts (e.g., 18.8–46.5 cases per 1000 person-years), with rates of 54.3–551.8 cases per 1000 person-years being reported [36,39,47,48,74,76,77]. These findings reinforce existing evidence indicating female military recruits are at increased risk of stress fracture during basic training and may be particularly exposed in highly physically demanding branches of service such as the Army. The highest incidence rate recorded for stress fractures was in an IDF cohort of anti-aircraft infantry recruits, which experienced an incidence rate of 1713 stress fractures per 1000 person-years of training exposure [66]. This high rate is reflective of the previously mentioned (and observed) high rates of stress fracture within the IDF [16,79], and may reflect specific characteristics of the IDF training environment or conscript selection.

It is important to note these observed rates of stress fracture occurrence within military recruit populations were considerably higher than those observed in qualified military personnel. For example, the overall incidence rate in active-duty cohorts of the U.S. armed forces ranged from 2.7 to 4.12 stress fractures per 1000 person-years [10,35,56]; however, similar to the situation for recruits, the incidence rate in qualified IDF soldiers was higher than the rates in U.S. military personnel, with a rate of 56.8 stress fractures per 1000 person-years [23]. This may again reflect contextual and methodological differences, though what these might be remains to be elucidated. Further, within a cohort of deployed U.S. military personnel, the incidence of stress fractures was even lower, at just 0.7 stress fractures per 1000 person-years [10], with reasons again unclear, though it might be hypothesised deployed personnel had a higher average level of fitness and health due to having met requirements for deployment. Overall, the available evidence clearly indicates the incidence rate of stress fractures is that orders of magnitude greater in recruits than in qualified personnel. As is the case with other common musculoskeletal conditions, the differences between the incidence rates in the two populations is likely a result of the combination of physical activity and medical histories (i.e., prior fitness for military service), natural or self-selection, different levels of personal control each group have over their workloads, and the training tasks involved in basic training regimes—involving large volumes of marching, walking, and running-based activities in which physical workloads are externally controlled [11]. Recruits are often physically and mentally unprepared for the military-related activities which are common in basic training regimes, and therefore are likely susceptible to overuse-related musculoskeletal injuries like stress fractures [11].

Overall, the existing evidence across differing military forces and personnel groups indicates the largest proportions of stress fractures occur at the tibia, fibula, and/or metatarsals [10,12,20,26,57,69,81]. Stress fractures at the talus, femur, femoral neck, or pelvis were reported, but less common [52,53,55,56,68,78]. The evidence from studies considered in the current review generally indicated there were limited differences in the anatomical distributions of stress fractures between male and female military personnel or recruits. However, Kardouni and colleagues [64] did observe a higher proportion of stress fractures affecting male recruits occurred at the tibia/fibula and metatarsals when compared to female recruits, despite a higher overall incidence rate for stress fractures within the female cohort. Consistent with this, female Finnish military conscripts experienced stress fractures of the pelvis or femoral bone at 8.2 times the rate at which these types of stress fracture were suffered by male conscripts [78]. Further research should seek to compare the specific sites of stress fractures in different military cohorts to determine if there are consistent and meaningful variations in the anatomical distributions of stress fracture between male and female recruits and cohorts defined in other ways (for example, by service arm or rank). Overall, similar anatomical distributions of stress fractures to those reported here for male recruits have also been described in athletic populations, and, in particular, runners, with higher rates of stress fractures affecting the tibia than other sites reported [1,14].

Comparisons of stress fracture incidence rates between military and civilian populations are challenging due to the limited investigation of stress fracture incidence outside specific athletic groups [100]. The existing literature predominantly focuses on athletic and military populations, where stress fractures are recognised as a significant concern. Stress fractures account for approximately 20% of presentations to sports medicine clinics, yet data on their incidence within athletic populations remain sparse [1]. Reported incidence rates in U.S. collegiate athletes range from 10.1 to 11.6 cases per 1000 athlete-years [98,99], with higher rates observed in endurance sports such as cross-country running (up to 53.5 cases per 1000 athlete-years) than in team sports like basketball (up to 35.7 cases per 1000 athlete-years) [98,99]. Despite these findings, military recruits appear to exhibit even higher stress fracture rates, suggesting unique risk factors inherent to military training environments.

The elevated incidence of stress fractures in military recruits, compared to athletes, may be attributed to contextual factors such as lower average baseline physical fitness, non-specific training regimens, and the demanding physical requirements of military training, including load carriage, weaponry, and prolonged activity in specialised footwear and body armour. While athletic training often incorporates individualised and periodised programmes aimed at injury prevention and performance enhancement, military training frequently adopts uniform, high-volume approaches. Although empirical data on stress fracture rates in broader civilian populations are lacking, the intense physical demands of military service likely contribute to higher stress fracture incidence rates than those likely experienced in most civilian occupations. Further research is warranted to substantiate these assumptions and explore stress fracture epidemiology across diverse populations.

### 4.2. Stress Fracture Risk Factors

Specific risk factors identified within this review for stress fractures in military personnel included a number of intrinsic risk factors, some related to recruit demographics. Consistent with evidence for many other types of musculoskeletal injuries [101,102], risk of developing a stress fracture was identified to be greater where there was a prior history of stress fracture, in cohorts of Chinese military recruits [20] and USMA cadets [75]. Prevention of primary injury is therefore potentially of paramount importance, and recruits with a history of stress fracture prior to entry to basic training should be carefully monitored. Additionally, this study found older military recruits and qualified personnel were at an increased risk of stress fracture, regardless of service branch [10,27,36,44,55,56,77,78]. This age risk factor was observed consistently across multiple nations, and incidence of stress fractures progressively increased as the age of personnel increased from the lowest age bracket (<20 years), through to the age bracket 40 years or older.

Stress fracture risk was also higher in both female military recruits and female qualified military personnel, when compared to male counterparts [10,35,39,44,47,48,56,66,68,72,78,93,95]. Most large-scale cohort studies of military recruits and qualified personnel from varying nations and services indicated the odds of stress fracture for female compared to male personnel were 3–8 times higher [35,44,47,56,78].

Additional female-specific risk factors were also identified and investigated within this review. For example, several variables relating to menstruation and birth control indicated younger menstruating females were at an increased risk of stress fracture than older females [68], and female recruits who were diagnosed with amenorrhea or reported no menstrual periods in the previous 12 months had increased odds of stress fracture [27,84]. More positively, the use of depot medroxyprogesterone acetate (a form of birth control medication) was not clearly associated with risk of stress fracture [27]. These results suggest that relative energy deficiency (RED), of which a common sign is amenorrhea [103], is occurring in some female recruits and may be contributing to their stress fracture risk. The importance of considering this factor (i.e., RED) is well known in sports (RED-S [104]) and is gaining attention in the military (RED-M) [105]. The findings indicate defence agencies should be providing nutritional and medical care and information to female recruits, along with guidance about monitoring menstruation and presenting for health care if amenorrheic, so that associated risks, including risks of relative energy deficiency and stress fracture, can be mitigated.

Other personnel characteristics that have been evaluated as possible risk factors for stress fractures in military personnel [1,14] included body composition, smoking status, alcohol consumption, and height. In this review, the associations between many of these variables and risk of stress fracture development were unclear (with confidence intervals often crossing 1.00) and the associations identified were sometimes conflicting. For example, there was limited overall evidence supporting a clear association between smoking history and stress fractures. However, when accounting for the entirety of the evidence evaluated (including point estimates and widths of the confidence intervals) there may be a marginal hazardous association for this variable [20,27,36,76,77]. Nevertheless, there is considerable uncertainty in these findings, with the 95% confidence intervals around the resulting estimates from some of the studies in question crossing 1.00 [27,36,76]. Similarly, there were unclear associations between heights of military recruits and stress fracture risk [20,78,81]. However, previous individual studies of military populations have found associations between the two variables, perhaps suggesting contextual differences may play a role in the association, where it exists [106,107]. Conversely, there were strong relationships identified by included studies, between alcohol consumption of greater than ten drinks per week and increased risk of stress fracture in female Army recruits [76,77]. The risks of long-term excessive alcohol intake are well understood, with such alcohol intake associated with increased risk of fragility fractures and low bone mass in both males and females, with the risk higher in men than in women [106]. This is particularly important to highlight, given the modifiable nature of this risk factor (i.e., alcohol intake).

The other commonly evaluated recruit characteristic was BMI or body composition, which was assessed for its association with stress fractures. However, the estimated comparative risk levels associated with varying categories of BMI or body composition measures in the available studies are difficult to interpret. This is largely due to different approaches used to dichotomise these variables, and whether the cohort was split by sex or considered as a whole. Overall, evidence indicates military recruits who are in the ‘underweight’ BMI category (<18.5 kg/m^2^) are at greater risk of developing stress fractures than those in the ‘normal’ (i.e., 18.5–24.9 kg/m^2^) or overweight/obese (>25 kg/m^2^) BMI categories [36,44,45]. Less clear is the association between the overweight or obese categories of BMI and stress fracture risk; although based on the available evidence, recruits in these classifications do not seem to be at increased risk [36,44,78]. However, it is important to note that the body composition literature varies widely in methodological and analytical approaches, and this presents a considerable limitation when seeking to interpret the estimated associations of these variables with stress fracture risk. Future research should evaluate the impacts of varying cut points for BMI on risk estimates, seek to establish methodological guidelines, and consider and control the effects on risk estimates of combining or splitting cohort-based findings by sex, within military populations.

The incidence of stress fractures was typically higher in the initial stages of military recruit training [51,64,69,78], with reductions in incidence observed in some contexts following the initial 3–4 months [51,69,78], and from 9 to 26 weeks after service commencement in U.S. Army soldiers [64]. Kardouni et al. [64] suggested this initial period is a time when recruits are initiating a new training regime and/or are increasing weight-bearing or impact types of activities. They also noted the associated increase in repetitive loading of bone tissue is consistent with current understanding of the pathophysiology of stress fractures, in which such repetitive loading may lead to bone fatigue damage, increased bone remodelling with impacts on bone integrity, and ultimately fracture. This, in turn, suggests training volume may be an important risk factor for stress fracture development. Of the studies which sought to quantitatively examine associations between training volume and stress fractures, observational data from Moran et al. [79] supported such a relationship, while a quasi-experimental trial by Popovich et al. [92] identified no significant associations. However, both studies were limited by a lack of randomisation and their findings may have been confounded by several variables. As such, the evaluated evidence, which largely consists of observational study designs, makes it difficult to identify unbiased estimates of the levels of stress fracture risk associated with training volume. In another cohort study of IDF recruits, Milgrom et al. [46] also did not find any significant relationships between variables relating to external training load (i.e., terrain changes, reductions in marching, etc.) and stress fracture risk [46]. However, restricting the *actual* training completed to the *planned* training prescription was associated with reduced stress fracture rates [46]. This has previously been highlighted as an issue [46,65], where platoon commanders may be deviating from planned training session volumes or intensities, potentially leading to increases in injury risk. While this study [46] was limited by a lack of control over confounding, it does again suggest an association may be present between completed training volume and incidence of stress fractures.

Overall, these results indicate there is a requirement for further higher-quality studies, of appropriate designs, to elucidate relationships between training volumes and intensities, and stress fracture risk in military populations. Such studies also have potential to identify the relative contributions to stress fracture risk of other modifiable and non-modifiable risk factors identified in this review, when considered alongside these training variables. Finally, of note were the findings of one study [90] indicating that, when wearing a specially designed fighting vest fitted specifically for the female anatomy rather than the standard body armour previously worn by personnel of both sexes, female recruits from a light infantry unit had a slightly increased incidence of stress fracture. This likely occurred since, even though the armour was better fitted, it was slightly heavier (1350 vs. 1950 g) than the conventional fighting vest [90]. Defence agencies need to be aware of this load issue as a potential concern in female recruits, who are likely to have lower initial levels of physical preparedness for military specific training when compared to males [108]. Reductions in initial loading, or implementation of prior training programmes, may need to be considered to reduce the impacts of loading on stress fracture incidence in female military recruits.

Unfortunately, there are no data available to quantify associations between physical training within the military occupational environment and stress fractures in qualified military personnel. However, the findings discussed above, in relation to recruit cohorts, are indicative of a protective relationship between higher levels of initial fitness and stress fracture risk. Once physiological and physical adaptations have taken place during initial stages of military training, a reduction in the incidence of stress fractures appears to follow in subsequent periods of military service. Overall, these findings suggest there may be no clear association between military service and stress fracture risk in qualified personnel, unless there is a period where there is a reduction in physical workload volume or intensity that results in a detraining or deconditioning effect on qualified personnel, or a period in which physical workloads substantially escalate to unaccustomed levels that cause bone fatigue. Such periods may occur during leave of absence, change in duties, or following rest after an injury. All of these factors should be taken into account when reintegrating qualified personnel back into regular training regimes or substantially increasing their physical workloads, where they may initially be at increased risk. Further research should be conducted to evaluate whether these hypothesised mechanisms of heightened risk in qualified personnel are of concern.

Connected to the physical training conducted as part of normal military duties, some included studies investigated whether prior individual physical activity history or physical performance characteristics moderated the stress fracture risk profile of military recruits. Overall, there was a consistent protective association observed for military recruits with a substantial history of exercise prior to entry into basic training regimes [20,27,51,68,77,80]. However, the available evidence was unclear regarding the specific training dose required in prior exercise for this protective relationship to exist [20,27,51,68,77,80]. Some studies suggested two to three or more exercise sessions/week in the previous 12 month period [27,51], or ≥seven hours of exercise per week were associated with a reduced risk but whether or not lower levels of exercise might also be protective was not clear [20,68]. These results demonstrate future research should seek to determine the minimum effective training dose or prior exercise history that is required to observe a meaningful protective effect against stress fracture development during subsequent periods of military basic training. For example, the predictive model developed by Moran et al. [80] indicated training volume (i.e., mins per session) and training frequency (i.e., sessions per week) were the predominant predictor variables for stress fracture in IDF recruits. This was likely an observation of the relationship between an individuals’ ‘fitness’ and corresponding ‘fatigue’ response to an exercise session [109,110], and demonstrated that a prior history of exercise involving shorter session durations with higher frequencies was associated with lower risk of stress fractures in IDF recruits [16]. These findings are likely supportive of mechanical load adaptation models, which would propose a recruit’s muscular, tendinous, and skeletal bone structures require training programmes which are inclusive of both appropriate loading periods and adequate rest periods, for bone and soft tissue remodelling to safely occur. Further research on these observed associations may allow the development of evidence-based guidelines for training prior to entry to the military training environment.

It should also be noted that the higher risk of stress fractures for female military recruits discussed previously in this report may be a proxy whereby sex per se may not increase the level of risk, but is instead a proxy indicator of average aerobic fitness capacities and stature [111]. As demonstrated in this report, there are associations between aerobic fitness and prior exercise history and risk of stress fractures, which have also been reported elsewhere within the literature [112,113]. Future research should seek to conduct appropriately designed studies to review the effect of potentially confounding factors such as aerobic fitness and prior exercise history in recruits of female sex specifically, in order to better understand the impact of factors related to biological sex.

Conversely, however, there was more limited evidence for a further protective benefit against stress fractures when considering higher levels (as opposed to presence) of aerobic fitness or muscular strength [20,24,27,78,96]. Overall, the available evidence suggested there may be a threshold level of aerobic performance that is associated with lower risk of developing stress fractures, with this association between aerobic performance and stress fractures not being dose-dependent [36,74]. This suggests the apparent protective benefits of fitness may be largely attributable to associated physiological training adaptions resulting from prior mechanical loading of the appropriate biological tissues (i.e., muscle, bone, tendons, etc.) in preparation for subsequent military-specific training involving similar frequencies, durations, and levels of loading. Correspondingly, increased or elevated aerobic performance beyond this point may not be associated with substantial further risk reductions. However, it is important to note some evidence did suggest further protective benefits, with these likely being less substantial.

Overall, the findings of included studies largely support the theorised mechanical load-tissue response aetiology model for stress fracture [9]. For example, the evidence regarding the incidence and timing of stress fracture occurrence during recruit training is largely in accordance with this theoretical model, and so is the finding that higher stress fracture incidence rates were largely associated with greater volumes of training [80]. However, there are a multitude of factors that contribute to tissue stress and strain, overload, and injury; the key to harnessing this knowledge to design preventive interventions will be the ability to account for and effectively monitor the physiological and biomechanical stress and strain induced by military training. Of importance in such efforts will be the development of new monitoring technologies to continually assess external training loads and so estimate associated stress and strain forces, numbers of mechanical load cycles, and the effects these may have on tissue breakdown and failure [9].

Accordingly, defence agencies may consider implementing, as an initial step, guidelines or pre-requisite levels of training prior to recruitment (i.e., pre-recruitment), which extend beyond aerobic performance benchmarks to encompass thresholds for prior training frequency and volume. Further work on appropriate implementation of contemporary training principles, like progressive overload, may also be useful to reduce or minimise incidence of stress fractures in military recruits, and outcomes of this work should be evaluated to guide and refine subsequent efforts. Additionally, the inclusion of athletic trainers embedded within a cohort of Air Force recruits assisted in reducing the incidence of stress fractures in one included study [89], and this could be another consideration for defence agencies.

Several additional risk factors that may mediate the relationship between stress fracture risk and individual recruit characteristics, military training, and prior exercise history (as a proxy for overall physical preparedness) were identified; these included pharmacological, genetic, and other blood-based biomarkers. For example, the evidence suggested there may be a threshold level of 25(OH)D which is associated with reduced risk of stress fractures [24,70], although the precise level was unclear, as the studies evaluated markedly different levels. This finding is largely in accordance with a meta-analysis on 25(OH)D serum concentrations as a risk factor for stress fractures, from which results indicated a protective effect of higher concentrations, with higher concentrations present more in controls than in stress fracture cases [114]. Again, precise concentrations of serum 25(OH)D associated with reduced risk were unclear, and as a general recommendation defence agencies may want to ensure military recruits have serum levels of 25(OH)D not considered clinically to be ‘low’ [114]. Supplementation with vitamin D and calcium was also found to be associated with a reduced stress fracture risk in military recruits [27].

Lower levels of iron, iron deficiency anaemia, and intake of NSAIDs were each also consistently associated with increased risk of stress fractures in military recruits [22,23,25,26,81]. In addition, Zhao and colleagues [20] reported several genetic markers were significantly associated with an increased risk of stress fracture, and defence agencies could consider whether these are risk factors they may wish to analyse in future recruit cohorts. There were also several biomechanical risk factors for stress fracture identified (e.g., high foot arch, subtalar joint kinematics, and use of shock-absorbing insoles), though the evidence for these was limited. Again, defence agencies could consider including these sorts of factors (excluding genetic risk factors) in pre-employment screening designed to reduce incidence rates of stress fracture [41,71,73]. This might be viable if these factors were confirmed to be significant risk factors in further, rigorous research, and if applicant pools were sufficiently large to allow for exclusion of individuals who were identified in these ways to be at heightened risk. However, casting doubt on some of these findings, a recent meta-analysis identified that footwear and orthoses did not have any clear effects on injury outcomes in military personnel [115]. Other markers or interventions for which there was unclear or contested evidence of associations with stress fracture risk included baseline parathyroid hormone levels [70], TRACP-5b (i.e., serum tartrate acid phosphatase isoform 5b) levels [83], deoxypyridinoline (i.e., a biomarker of bone resorption), ferritin [25,81], IL-6 [25], C-reactive protein [25], and prenatal vitamin supplementation [116]. Consequently, based on the available evidence, the allocation of resources to interventions based on addressing these factors may need to be reconsidered by defence agencies.

### 4.3. Limitations

It was challenging to ascertain whether incidence rates reported in the included studies were *fracture* incidence rates or *case* incidence rates, since included studies often provided scant detail on exactly how incidence rates were calculated and used terms like ‘case’ in various ways. This highlights the limitations in synthesising a broad evidence base derived from heterogenous samples, study designs, and methods and it is noteworthy there was little commentary or analysis in the included studies of potential explanatory variables such as these. In future, authors of primary research should seek to add, and where possible, adjust analyses for additional contextual details and potential explanatory variables which may assist in identifying specific reasons (or contribute to hypotheses) for variability. Authors should also ensure they provide clear definitions of terms like ‘case’ and sufficient details of the methods used to estimate incidence rates.

Formal inter-rater reliability statistics were not calculated for the article screening processes; instead, reviewer discrepancies were resolved through discussion to achieve consensus. While this approach ensured conceptual consistency and timeliness, it may limit the reproducibility of reviewer agreement. Further, a limitation of this review is the potential for confounding arising from the underlying study designs within the evidence base. While many prospective cohort studies have reported incidence rates of stress fractures in military populations, relatively few high-quality studies have thoroughly examined associated risk factors. Consequently, analyses with limited adjustment for explanatory variables reduces the confidence with which specific exposures or preventive interventions can be linked to stress fracture occurrence.

Additionally, despite the considerable volume of overall evidence relating to incidence rates and risk factors, there was a general lack of high-quality data regarding *specific* threshold levels of exposure to occupational *tasks* as *risk factors* for *stress fractures*. Consequently, there was largely only observation-level data relating stress fracture risk to training load exposures. Issues of dichotomisation or categorisation of risk factor variables were also present in many of the included studies [36,37,74,78]. Briefly, while there might be some advantages relating to simplicity for dichotomising or categorising continuous variables (e.g., data visualisation), the costs of doing so include reduction in statistical power, increasing risks of false positive or negative findings, and being forced to use arbitrary cut-points for data. These issues have been explained and the implications considered in depth elsewhere [117,118].

## 5. Conclusions

The findings from this systematic review demonstrate military recruits are at substantial risk for developing stress fractures. Specific sub-populations within the military were at particular risk, including recruits undertaking initial training, older personnel, female service members, female members consuming more than ten alcoholic drinks per week, underweight service members, recruits entering basic training without a prior history of exercise involving a frequency of ≥2 times/week or ≥7 h per week, recruits with low serum 25(OH)D (vitamin D) levels, and recruits with lower levels of iron or iron deficiency anaemia, or intake of NSAIDs. Recruits were also at greatest risk of stress fracture during the initial stages of military training (e.g., initial 3–6 months), and at stages of physical training involving the highest volumes of overall loading. Interestingly, increased aerobic performance was preventative up to a threshold level, with higher performance not always indicative of increased protection from stress fractures; this indicates the relationship may not be dose dependent. Preventative strategies to reduce stress fracture risk were difficult to elucidate; however, there was limited evidence suggesting that restricting physical training to the *planned* load and prescription (i.e., avoiding additional, unplanned physical activity) was linked with a reduced incidence of stress fractures, and the use of shock-absorbing insoles and employment of an athletic trainer to care for military cohorts also reduced the incidence. Further high-quality, large-scale, prospective cohort studies and experimental trials should be conducted to better understand the mechanisms involved in modifying risk. Such studies may support the development and testing of specific preventive interventions and elucidate exposure thresholds associated with heightened risk of fracture (e.g., through utilising wearables such as GPS or accelerometry).

## Figures and Tables

**Figure 1 ijerph-22-01760-f001:**
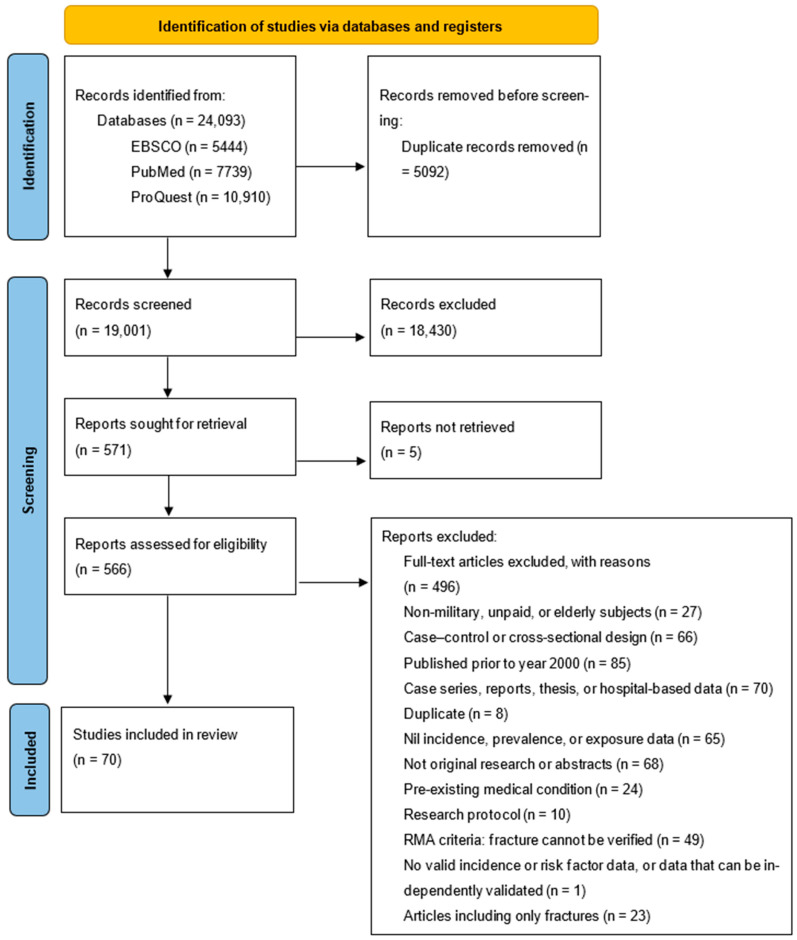
PRISMA diagram showing results of the search, screening, and selection processes [28].

**Table 1 ijerph-22-01760-t001:** An example of the search string used in PubMed—a title/abstract filter was used.

Database	Search Terms	Filters (PubMed)
PubMed	(risk [Title/Abstract] OR predict * [Title/Abstract] OR prevalence [Title/Abstract] OR incidence [Title/Abstract] OR caus * [Title/Abstract] OR etiol * [Title/Abstract] OR frequenc * [Title/Abstract] OR rate * [Title/Abstract] OR mediat * [Title/Abstract] OR exposure * [Title/Abstract] OR likelihood [Title/Abstract] OR probability [Title/Abstract] OR factor [Title/Abstract] OR factors [Title/Abstract] OR hazard [Title/Abstract] OR hazards [Title/Abstract] OR predisposing [Title/Abstract]) AND (work * [Title/Abstract] OR occupation * [Title/Abstract] OR profession * [Title/Abstract] OR trade [Title/Abstract] OR employ * [Title/Abstract] OR military [Title/Abstract] OR Defence [Title/Abstract] OR Defense [Title/Abstract] OR airforce [Title/Abstract] OR “air force” [Title/Abstract] OR army [Title/Abstract] OR navy [Title/Abstract] OR recruit [Title/Abstract] OR soldier * [Title/Abstract] OR marines [Title/Abstract] OR “Military Personnel” [Title/Abstract]) AND (Fracture * [Title/Abstract] OR stress fracture * [Title/Abstract] OR overuse fracture * [Title/Abstract] OR bone stress * [Title/Abstract] OR bone strain * [Title/Abstract])	English, Portuguese, Italian, Spanish Languages, Humans

**Table 2 ijerph-22-01760-t002:** Inclusion and exclusion criteria.

Inclusion	Exclusion
Studies reporting original quantitative research conducted in humans aged 16 years or older, in which cohorts of participants were followed over time in a longitudinal study design (for example, cohort studies, randomised controlled trials, quasi-experimental studies);	Literature reviews of any type, cross sectional studies, case–control studies (except any nested within cohort studies);
Studies published in English, or translatable to English from Portuguese, Spanish, Italian, or French, by members of the research team;	Published abstracts;
Studies investigating factors, or exposures, or hazards, or causes, or mediators associated with development or prevention of stress fractures in personnel engaged in military occupations, or the incidence, or prevalence, or likelihood of the condition occurring in military occupational groups; and	Non-peer-reviewed articles and reports; Articles that are not reports of original research;
Studies using diagnostic criteria consistent with the criteria proposed by the Repatriation Medical Authority’s Statements of Principles for stress fractures, as follows:	Studies of pharmacologic interventions or ergogenic aids; or
Means an acquired break or rupture of bone resulting from fatigue of the bone; and Excludes spondylolysis, pathological fractures, periostitis, stress fractures due to insufficiency of the bone, and bone stress injuries/bone marrow oedema not being a stress fracture.	Studies of unpaid elite athletes, volunteer occupations, or non-military occupations.

**Table 3 ijerph-22-01760-t003:** Military recruits lower extremity stress fracture incidence rate and IRR.

Military	Branch	Male Incidence Rate * (Stress Fracture; Case-Based)	Female Incidence Rate * (Stress Fracture; Case-Based)	Combined Incidence Rate *	Incidence Rate Ratio (Female: Male)
United States	Army	27.42–311.4 [39,64,92]; 25.6–46.5 [39,48]	106.66–871.7 [39,64,76,77]; 54.3–551.8 [27,36,39,48,74,76,77]	40.38–158.6 [39,44,64]; 29.2–158.6 [39,44,47,48]	2.14–4.6 [35,39,44,48,64,82]
	Marine Corps	N/A; 81.3–270.1 [2,63,93]	244.4–365 [13,63,84]; 182.9 [93]	126.4 [50,63]; N/A	2.25 (95% CI, 0.28–18.3) [93]
	Air Force			N/A; 87.6–105.9 [89]	
	Navy		N/A; 383.5–540 [27,74]	N/A; 34.9 [60]	
	Cadets	19.9 [68]; 14.2 [68]	69.9 [68]; 47.8 [68]	27.5 [68]; 19.3 [68]	
	Military *	29.6 [10]; N/A	94.7 [10]; 552 [77]	44.2 [10]; N/A	
Israel	Infantry and Army	N/A; 300–810.5 [16,73,91]	N/A; 245.1–369.9 [26,81]		8.93 (95% CI, 2.22–35.90) [91]
	Basic and Advanced Training	N/A; 727.2 [79]	N/A; 309 [25]	N/A; 142.2 [43]	
	Anti-aircraft training	1248 [66]; 582 [66]	2976 [66]; 1243 [66]	1713 [66]; 760 [66]	2.83 (95% CI, 1.89–3.00) [66]
	Elite combat unit	517.2 [80]; 387.9 [80]			
Australia	Basic training	13.7 [49]; N/A	59.3 [49]; N/A	17 [49]; N/A	4.41 (95% CI, 2.33–8.35) [49]
Finland	Basic training	15.2 [51]; 83.8–116 [24,85]		N/A; 99 [83]	
United Kingdom	Army	114.6 [40]; 62.6–194 [12,86,87]	N/A; 163.9 [86]	N/A; 242.8 [86]	
	Royal Marines	63.4–202.3 [41,57,65,70]; 117.1 [70]			
India	Basic training	N/A; 87.3 [69]			
China	Basic infantry	N/A; 878.8 [20]			
South Africa	Basic training	0 [94]; N/A			

* Incidence rates are reported as follows: per 1000 person-years.

**Table 4 ijerph-22-01760-t004:** Stress fracture incidence rates in military recruits by anatomical location.

ANATOMICAL LOCATION	POPULATION	STRESS FRACTURE INCIDENCE RATE (PER 1000 PERSON-YEARS)
**TIBIA/FIBULA**	U.S. Military ♂♀	15.7 *stress fractures* [10]
**TIBIA**	U.S. Marine Corps ♀	200.8 *stress fractures* [13]
	British Army ♂	27.8 *cases* [12]
	British Royal Marines ♂	20.2 and 26.3 *cases* [57]
	Finnish Conscripts ♂	22.3 *cases* [85]
	Chinese Infantry ♂	293 *cases* [20]
**METATARSALS**	U.S. Military ♂♀	5.6 *cases* [10]
	British Army ♂	24.8 *cases* [12]
	British Royal Marines ♂	11.4–68.9 *stress fractures* [41,57,71]
	Finnish Conscripts ♂	55.9 *cases* [85]
	Chinese Infantry ♂	447 *cases* [20]
**HIP/PELVIS**	U.S. Marine Corps ♀	54.8 *stress fractures* [13]
**PELVIS**	U.S. Marine Corps ♀ U.S. Military ♂♀	52.8 *stress fractures* [84] 1.4 *stress fractures* [10]
**FEMUR**	U.S. Marine Corps ♀	48.6 *stress fractures* [84]
	British Army ♂	5.2 *cases* [12]
	British Royal Marines ♂	8.1 *stress fractures* [57]
	Chinese Infantry ♂	41.6 *cases* [20]
**PELVIS/FEMUR**	Finnish Conscripts ♂	3.1 *stress fractures* [78]
	Chinese Infantry ♂	60.5 *cases* [20]
**TALUS**	Finnish Conscripts ♂♀	0.09 *stress fractures* [55]
**CALCANEUS**	Finnish Conscripts ♂	5.6 *cases* [85]
	British Army ♂	4.8 *cases* [12]
**FIBULA**	British Royal Marines ♂	2.3 *stress fractures* [57]
**FEMORAL NECK**	Chinese Infantry ♂	37.1 *cases* [20]
	U.S. Military ♂♀	1.1 *stress fractures* [10]
**DISPLACED FEMORAL NECK**	British Royal Marines ♂	1.51 *stress fractures* [38]
	Finnish Conscripts ♂	0.023–0.053 *stress fractures* [53]
**UNDISPLACED FEMORAL NECK**	Finnish Conscripts ♂	0.132–0.532 *stress fractures* [52]
**DISPLACED FEMORAL SHAFT**	Finnish Conscripts ♂♀	0.015 *cases* [54]
**FEMORAL SHAFT**	U.S. Military ♂♀	0.7 *stress fractures* [10]

♂: Male recruits; ♀: Female recruits; U.S.: United States.

## Data Availability

The original contributions presented in this study are included in the article/Appendix A. Further inquiries can be directed to the corresponding author.

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
