# Peer review of "Incidence and Risk Factors for the Development of Stress Fractures in Military Recruits and Qualified Personnel: A Systematic Review"

_ijerph, 2025, doi:10.3390/ijerph22111760_

Round 1
Reviewer 1 Report
Comments and Suggestions for Authors
Thank you for the opportunity to review this paper. I found this an interesting and important read. I also found it to be very long. I do not know if that is an issue for the editors.
Methods:
Provide the inter-rater reliability info for the two authors extracting the results (and any third party involved or list as a limitation of the review).
Although mentioned, there is also some confusion at times with "case based" and "fracture based" incidence rates which muddies the interpretation and interpretation somewhat.
Also, the review inadequately addresses confounding in observational studies. Most included studies fail to adjust for multiple risk factors simultaneously. It fails to systematically evaluate whether the studies employed applied appropriate multivariate analysis.
The evidence regarding training volume remains unexplored despite its theoretical importance. Careful discussion of study designs and populations might assist this issue.
I believe the results could be condensed and still provide similar overall content. Also, the discussion often repeats content from the results. That too could be streamlined to reduce length.
Otherwise good paper. Publish!
Author Response
Comment 1: Thank you for the opportunity to review this paper. I found this an interesting and important read. I also found it to be very long. I do not know if that is an issue for the editors.
Response: We thank the reviewer for their assessment of our manuscript. We agree that the review offers a comprehensive synthesis of evidence relating to a critical type of MSKI in military populations, and the length of the review reflects this. However, we believe the review in its current comprehensive form is an extremely valuable evidence-base on stress fractures in military populations which will inform risk mitigation of this injury type to both researchers and practitioners.
Comment 2: Methods: Provide the inter-rater reliability info for the two authors extracting the results (and any third party involved or list as a limitation of the review).
Response: We thank the reviewer for this helpful comment and agree that including a quantitative measure of inter-rater reliability (e.g., Cohen’s kappa) would have strengthened the methodological reporting. In this review, screening was conducted by two authors, with discrepancies resolved through discussion to achieve consensus. Given the conceptual nature of several extracted variables and the need to expedite the review for grant-related timelines, we did not calculate a formal inter-rater reliability coefficient. Instead, disagreements were addressed through iterative discussion, often via video conference, until full agreement was reached. This has now been acknowledged as a limitation within the manuscript (see lines 1084–1087). These lines read: "Formal inter-rater reliability statistics were not calculated for the article screening processes; instead, reviewer discrepancies were resolved through discussion to achieve consensus. While this approach ensured conceptual consistency and timeliness, it may limit the reproducibility of reviewer agreement."
Comment: Although mentioned, there is also some confusion at times with "case based" and "fracture based" incidence rates which muddies the interpretation and interpretation somewhat.
Response: We thank the reviewer for highlighting this important issue. We agree that inconsistent use of the terms “case-based” and “fracture-based” incidence rates across the included studies creates challenges for interpretation and synthesis. As noted in the limitations section, the published literature often applies inconsistent or ambiguous definitions of incidence rates whether they refer to case incidence (i.e., the number of individuals sustaining stress fractures) or fracture incidence (i.e., the total number of stress fractures observed). To address this, we clarified this issue within the manuscript and, where possible, calculated and reported both types of incidence rates to ensure the most accurate representation of the data. We have also emphasised in the limitations that this lack of definitional clarity restricts comparability between studies. Future research should provide clear operational definitions and detailed methodological descriptions to improve the interpretability and synthesis of incidence data in this field.
Comment: Also, the review inadequately addresses confounding in observational studies. Most included studies fail to adjust for multiple risk factors simultaneously. It fails to systematically evaluate whether the studies employed applied appropriate multivariate analysis.
Response: We thank the reviewer for this important observation. We agree that confounding is a key methodological limitation across the available literature. The review already addresses this issue in several sections, noting the predominance of observational designs and the limited use of multivariate adjustment within included studies. The extracted data tables (in supplemental material 4 & 5) indicate whether analyses were univariate or multivariate, allowing readers to appraise the degree of adjustment employed. To clarify this point, we have revised the text in the Discussion and Limitations sections to make these issues more explicit and to emphasise how the lack of appropriate control for confounders limits the strength of causal inference in this field. Lines 1087-1093 now read "Further, a limitation of this review is the potential for confounding arising from the underlying study designs within the evidence base. While many prospective cohort studies have reported incidence rates of stress fractures in military populations, relatively few high-quality studies have thoroughly examined associated risk factors. Consequently, analyses with limited adjustment for explanatory variables reduces the confidence with which specific exposures or preventive interventions can be linked to stress fracture occurrence."
Comment: The evidence regarding training volume remains unexplored despite its theoretical importance. Careful discussion of study designs and populations might assist this issue.
Response: We thank the reviewer for highlighting the importance of training volume as a theoretical risk factor for stress fractures. We feel this aspect has been discussed in detail within the Results and Discussion sections of the manuscript (e.g., see lines 907–948), where we specifically evaluate evidence from Moran et al. [80], Popovich et al. [93], and Milgrom et al. [47] concerning training volume, load, and associated stress fracture risk. These sections also address the limitations of these studies, including their observational design, limited control for confounding, and variability in how training load was operationalised. Given that these discussions already synthesise the available evidence and contextual limitations, we believe the current presentation appropriately reflects the state of knowledge in this area and that no further modification is required.
Comment: I believe the results could be condensed and still provide similar overall content. Also, the discussion often repeats content from the results. That too could be streamlined to reduce length.
Response: We thank the reviewer for this constructive comment and appreciate the observation regarding the length and potential repetition within the manuscript. We have reviewed the results and discussion sections to ensure that each presents distinct and complementary content, with any unnecessary repetition reduced where possible. The primary aim of this review is to provide an authoritative evidence base to better understand how physical training, training-related variables, epidemiological factors, and service categories influence stress fracture risk in military personnel. Given the breadth and complexity of the available literature, the overall length of the manuscript is commensurate with the scope of the review and reflects the need to present a rigorous and comprehensive synthesis of incidence rates, risk factors, and methodological considerations. The length also affords opportunities to discuss the results in the context of the limitations of the evidence and study designs highlighted by the reviewer in previous comments.
Comment: Otherwise good paper. Publish!
Response: We thank the reviewer for their support of the manuscript.
Reviewer 2 Report
Comments and Suggestions for Authors
Overall: The purpose of this systematic review was to examine the differences in incidence rate and risk factors for stress fractures among military personnel. The authors engaged in an extensive review of the scientific literature and summarized a detailed list of potential risk factors and incident rates within sub-populations of the military - in particular, recruits and qualified personnel. Given the high rate of stress fractures within militaries around the world, an exhaustive review of the current state of scientific knowledge is useful for both researchers and practitioners within this sector, as well as military leadership and policy developers.
Line 121: Why were articles prior to the year 2000 excluded? Some context as to why could be helpful.
Discussion/Conclusions: Given the extensive nature of this systematic review, developing a summative table (or some kind of figure) outlining the general risk factors observed across the overall review would help the reader.
Author Response
Comments 1:
Overall: The purpose of this systematic review was to examine the differences in incidence rate and risk factors for stress fractures among military personnel. The authors engaged in an extensive review of the scientific literature and summarized a detailed list of potential risk factors and incident rates within sub-populations of the military - in particular, recruits and qualified personnel. Given the high rate of stress fractures within militaries around the world, an exhaustive review of the current state of scientific knowledge is useful for both researchers and practitioners within this sector, as well as military leadership and policy developers.
Response: We thank the reviewer for their thoughtful assessment of our manuscript. We agree that the review offers a comprehensive synthesis of evidence relating to a critical type of MSKI in military populations and will be valuable to both researchers and practitioners.
Comment:
Line 121: Why were articles prior to the year 2000 excluded? Some context as to why could be helpful.
Response:
We thank the reviewer for this question and the opportunity to clarify our rationale. While we agree that systematic reviews should ideally not be arbitrarily date-limited, we chose to include studies published from 2000 onwards to ensure a manageable scope (the review is already very large) and to focus on data reflective of contemporary training practices, medical diagnostics, and military population demographics.
Comment: Discussion/Conclusions: Given the extensive nature of this systematic review, developing a summative table (or some kind of figure) outlining the general risk factors observed across the overall review would help the reader.
Response: We thank the reviewer for their comment on this potential improvement for the manuscript. This is something the authors have considered in previous iterations of the manuscript. However, the comprehensive nature of the review (as the reviewer mentions) makes it difficult to construct such a table or figure and ensure the accuracy of the data presented. Specifically, the varied study designs, univariate or multivariate (Adjusted) analyses provided by the studies and whether or where confounders have been accounted for in the analysis or design of the study, across differing military sub-populations (e.g., rank, branch of service, nation, recruit, qualified personnel etc.) mean the risk of misrepresenting an identified risk factor within a summary table is a substantive risk. As such, to maintain accuracy and preserve the contextual nuances of each study, we elected to present the findings narratively by sub-section, ensuring that each risk factor was interpreted in the most appropriate methodological and population-specific context.